# Clinical Outcomes and Safety of Transcatheter Arterial Embolization in Patients with Traumatic or Spontaneous Psoas and Retroperitoneal Hemorrhage

**DOI:** 10.3390/jcm13113317

**Published:** 2024-06-04

**Authors:** Chang Hoon Oh, Soo Buem Cho, Hyeyoung Kwon

**Affiliations:** 1Department of Radiology, Ewha Womans Mokdong Hospital, College of Medicine, Ewha Womans University, Seoul 07985, Republic of Korea; 01352@ewha.ac.kr; 2Department of Radiology, Ewha Womans Seoul Hospital, College of Medicine, Ewha Womans University, Seoul 07804, Republic of Korea; 3Department of Radiology, Chungnam University Hospital, Chungnam University School of Medicine, Daejeon 35015, Republic of Korea; hyeyoungkwon@cnuh.ac.kr

**Keywords:** hemorrhage, transcatheter arterial embolization, trauma, retroperitoneum

## Abstract

**Background:** We aimed to assess the effectiveness and safety of transcatheter arterial embolization (TAE) in the management of spontaneous or traumatic psoas and/or retroperitoneal hemorrhage. **Methods:** This single-center retrospective study enrolled 36 patients who underwent TAE for the treatment of psoas and/or retroperitoneal hemorrhage between May 2016 and February 2024. **Results:** The patients’ mean age was 61.3 years. The spontaneous group (SG, 47.1%) showed higher rates of anticoagulation therapy use compared with the trauma group (TG, 15.8%) (*p* = 0.042). The TG (94.7%) demonstrated higher survival rates compared with the SG (64.7%; *p* = 0.023). Clinical failure was significantly associated with the liver cirrhosis (*p* = 0.001), prothrombin time (*p* = 0.004), and international normalized ratio (*p* = 0.007) in SG and pRBC transfusion (*p* = 0.008) in TG. Liver cirrhosis (OR (95% CI): 55.055 (2.439–1242.650), *p* = 0.012) was the only identified independent risk factor for primary clinical failure in the multivariate logistic regression analysis. **Conclusions:** TAE was a safe and effective treatment for psoas and/or retroperitoneal hemorrhage, regardless of the cause of bleeding. However, liver cirrhosis or the need for massive transfusion due to hemorrhage increased the risk of clinical failure and mortality, necessitating aggressive monitoring and treatment.

## 1. Introduction

Psoas and retroperitoneal hemorrhages are critical medical conditions characterized by hypotension, abdominal/flank tenderness, and sensorimotor deficits, often secondary to nerve compression by a retroperitoneal hematoma [1,2]. Spontaneous retroperitoneal hemorrhage (SRH) occurs without an incident cause and predominantly affects elderly patients receiving anticoagulation therapy. Despite its rarity, SRH is associated with a high mortality rate, reaching up to 22% as reported [3,4,5]. Patients with this condition often present with multiple comorbidities that complicate their management.

Additionally, retroperitoneal hemorrhage can result from traumatic or iatrogenic injuries [6]. The complexity of managing retroperitoneal hemorrhage following blunt trauma is often accompanied by multiple traumatic injuries. This underscores the need for a nuanced treatment approach that takes into account the unique etiologies and patient factors involved [7]. However, the transcatheter arterial embolization (TAE) as a minimally invasive endovascular treatment option has shifted the treatment paradigm, particularly for patients who exhibit hemodynamic instability despite receiving the aforementioned conservative measures [3,4,5,8,9,10].

Our study evaluated the effectiveness and safety of TAE in managing bleeding due to trauma or spontaneous events in patients with psoas and/or retroperitoneal space. By assessing the efficacy of TAE and determining the factors that affect clinical outcomes, this study aimed to provide significant insights into the development of optimal strategies for addressing this complex and potentially critical condition.

## 2. Materials and Methods

This retrospective single-center study was approved by the institutional review board of Ewha Womans University (IRB No. EUMC 2024-03-040), which waived the need for obtaining informed consent from the patients. 

### 2.1. Patient Population

The information system of the authors’ hospital was retrospectively reviewed between May 2016 and February 2024. All methods were performed in accordance with the relevant guidelines and regulations. A total of 36 consecutive patients (21 men and 15 women) with a mean age of 61.3 years (range: 16–89 years) who underwent TAE as a treatment for psoas or retroperitoneal hemorrhage were enrolled in this study. The patients’ electronic medical records and images were thoroughly examined. The decision to perform TAE was based on computed tomography (CT) findings and clinical and laboratory evidence of continued bleeding despite adequate conservative therapy. Patients aged 10–90 years who underwent TAE owing to a suspicion of active bleeding in the psoas or retroperitoneal space, with or without traumatic injury, were included in this study. The exclusion criteria were trauma patients whose main bleeding site was not in the psoas or retroperitoneal space, like the spleen or liver, and also excluded were those without TAE.

### 2.2. Procedure

The right common femoral artery was accessed under local anesthesia, and the main target vessels visualized as potential bleeding sources on CT were explored. In addition, all arteries within the same anatomical territory surrounding the hematoma were systematically catheterized to determine other bleeding arteries or blood flow through collaterals from the primary bleeding site. If the hematoma was located in the psoas muscle, selective catheterization and opacification of the iliolumbar and homolateral lumbar arteries were performed. A 1.7- or 1.9-F microcatheter was advanced into all identifiable target vessels, which were subsequently embolized. Embolic agents included polyvinyl alcohol (PVA; Contour; Boston Scientific, Marlborough, MA, USA), n-butyl cyanoacrylate (NBCA; Histoacryl; B. Braun, Melsungen, Germany), detachable vascular microcoils (Interlock; Boston Scientific, Marlborough, MA, USA and Concerto; Medtronic, Minneapolis, MN, USA), and gelatin sponge particles (GSPs; Nexsphere, Next Biomedical, Incheon, Republic of Korea) or their combinations. Embolic materials were selected intraoperatively on a case-by-case basis at the discretion of the interventional radiologist. Subsequently, arteriography after embolization was performed to confirm the successful occlusion of the target vessel with no further bleeding (Figure 1).

### 2.3. Study Endpoints

Technical success was defined as the complete occlusion of the bleeding vessel and the absence of visible signs of active bleeding, such as contrast media extravasation (CME) or pseudoaneurysm, on immediate post-embolization angiography. Additionally, in cases where CME or pseudoaneurysm was not clearly identifiable on angiography, occlusion was performed on the vessel supplying the same anatomical territory surrounding the hematoma. Primary clinical success was defined as the resolution of the signs and symptoms of bleeding with no requirement for interventional or surgical hemostasis. The complications were classified as either major or minor according to the Society of Interventional Radiology Standards of Practice Committee guidelines [11]. In the present study, the abnormal platelet count (<5 × 10^4^/μL) and/or international normalized ratio (INR > 1.5) indicated coagulopathy as these values require correction through the infusion of platelets or fresh frozen plasma [12,13].

### 2.4. Statistical Analysis

Categorical variables were compared using the chi-square test or Fisher’s exact test, as appropriate. Continuous variables were expressed as the mean ± standard deviation and were compared using Student’s *t*-test. A multivariate analysis was performed to identify independent factors using binomial logistic regression. All statistical analyses were performed using SPSS software version 20.0, with a *p* value of <0.05 indicating statistical significance.

## 3. Results

The baseline demographic and clinical data of the patients included in this study are outlined in Table 1. Among the study patients, 11 (30.6%) received anticoagulation therapy. On CT scans, CME was identified in 91.7% (33/36) of patients. Among these patients, 69.4% (25/36) showed active bleeding in the form of CME or pseudoaneurysm on angiography. Meanwhile, 52.8% of patients (19/36) experienced bleeding from multiple sites. Technical success was achieved in all patients, with a clinical success rate of 77.8% (28/36) and an overall mortality rate of 19.4% (7/36). In the present study, which compared two groups, the spontaneous group (SG) comprised 17 patients, while the trauma group (TG) comprised 19 patients. The use of oral anticoagulation therapy (*p* = 0.042), underlying conditions (*p* < 0.001), multiple bleeding sites (*p* = 0.047), average systolic blood pressure (*p* = 0.045), initial hemoglobin levels (*p* = 0.018), and mortality rate (*p* = 0.023) were found to be significantly different between the groups.

Notably, all five patients with liver cirrhosis experienced clinical failure, while seven patients receiving oral anticoagulation therapy achieved clinical success. Clinical failure occurred in 8 patients, of whom 62.5% (5/8) had underlying liver cirrhosis, while 20.0% (2/8) sustained trauma. Only one patient received anticoagulation therapy. The signs of active bleeding, such as CME or pseudoaneurysm, were observed in all patients on CT scans and in 75.0% (6/8) on angiography. Furthermore, 62.5% (5/8) of patients had bleeding from multiple vessels. The average pre-treatment hemoglobin level was 6.03 ± 0.97 g/dL, while the average number of packed red blood cell (pRBC) units transfused was 5.6 ± 2.8 units. Among the total of eight patients who experienced clinical failure, one patient with neurofibromatosis suffered hemorrhage after a collision. This patient underwent a second session of TAE due to unstable vital signs following the first TAE. During the second session, another arterial bleeding site that had not previously been identified was discovered and successfully embolized, leading to clinical improvement. The remaining seven patients (87.5%) expired (Table 2).

In 17 SG patients, 6 patients experienced clinical failure. Among the SG patients with liver cirrhosis, there were a total of six patients, five of whom experienced clinical failure (83.3%, *p* = 0.001). Additionally, prothrombin time (13.5 ± 1.8 vs. 23.4 ± 5.5, *p* = 0.004) and INR (1.18 ± 0.17 vs. 2.03 ± 0.51, *p* = 0.007) were statistically significant (Table 3). Among the 19 TG patients, there were 2 cases of clinical failure. Among the variables, only pRBC transfusion was significant (clinical success: 3.65 ± 1.87 vs. clinical failure: 8.00 ± 2.83; *p* = 0.008) (Table 4).

The univariate logistic regression analysis demonstrated that liver cirrhosis (OR (95% CI): 43.333 (3.712–505.806), *p* = 0.003), hemoglobin levels (OR (95% CI): 12.600 (1.350–117.570), *p* = 0.026), and pRBC transfusion (OR (95% CI): 1.459 (1.024–2.078), *p* = 0.037) were risk factors associated with primary clinical failure in patients with psoas and/or retroperitoneal hemorrhage. However, liver cirrhosis (OR (95% CI): 55.055 (2.439–1242.650), *p* = 0.012) was the only identified independent risk factor for primary clinical failure in the multivariate logistic regression analysis (Table 5). 

One patient developed motor weakness in the ipsilateral lower extremity after TAE. The patient experienced left spontaneous psoas hemorrhage and underwent TAE. Although active bleeding was not observed on angiography, superselective embolization was performed prophylactically using GSP in the left third and fourth lumbar arteries. However, the patient later developed weakness in the lower limbs on the same side. Subsequent magnetic resonance imaging (MRI) showed diffusion restriction from the lower thoracic spinal cord to the conus medullaris, suggesting spinal cord ischemia (Figure 2). Meanwhile, other patients only developed minor complications, which included mild fever (four patients, 11.1%). These symptoms improved following conservative treatment within 2–3 days postoperatively.

## 4. Discussion

The retroperitoneum is a highly vascular area with numerous collateral supplies to any given region [14]. Hence, accessing this area poses a significant challenge. Once the bleeding artery is controlled, collateral supply to the same territory may trigger new bleeding episodes. Moreover, surgical intervention in the retroperitoneal space can often escalate the risk of catastrophic hemorrhage due to the dissection and loss of the passive tamponade effect of a hematoma [15]. Compared with surgical methods, endovascular procedures, including the transcatheter embolization of the bleeding vessel, are safe, quick, and minimally invasive, and provide immediate treatment [16]. The univariate logistic regression analysis showed that pRBC transfusion was a statistically significant factor associated with clinical failure (OR (95% CI): 1.459 (1.024–2.078), *p* = 0.037). Previous studies have shown that a higher number of pRBC transfusions before embolization is significantly linked to poorer clinical outcomes and increased mortality within 30 days [17,18]. Patidar et al. suggested that the development of transfusion-related disseminated intravascular coagulopathy prior to the embolization procedure could be the most likely explanation for clinical failure associated with massive blood transfusion [18]. Research on the risk factors of clinical failure during TAE for traumatic and spontaneous psoas or retroperitoneal hemorrhages remains limited. Hence, further investigations with a larger cohort are warranted.

Among the 37 patients examined in our study, 36 achieved technical success, with clinical success observed in 28 patients (77.8%). Notably, the clinical success rate was lower in the SG (64.7% (11/17)). The overall mortality rate was 19.4% (7/36), which is consistent with the reports of previous studies that documented mortality rates ranging from 12% to 22% with similar embolization outcomes [3,4,6]. However, the SG exhibited a significantly higher mortality rate of 35.3% (6/17), while the TG demonstrated a mortality rate of only 5.3% (1/19) (*p* = 0.023), surpassing previously reported figures. This increase in the mortality rate in the SG was attributed to the presence of liver cirrhosis in six patients in our cohort, five of whom experienced spontaneous hemorrhage, clinical failure, and eventually death within days. This outcome starkly aligns with the findings of previous studies, which reported a mortality rate between 71% and 83% in such patients, underscoring a poor prognosis [19]. Patients with liver cirrhosis are predisposed to bleeding due to the occurrence of complications such as thrombocytopenia, platelet dysfunction, and decreased levels of coagulation factors. In our study, the multivariate logistic regression analysis showed that liver cirrhosis was the only independent factor associated with clinical failure (OR (95% CI): 55.055 (2.439–1242.650), *p* = 0.012). Gastrointestinal hemorrhage is the most prevalent bleeding complication, although a significant association between retroperitoneal hemorrhage and alcoholic liver disease has been documented [19]. This association further exacerbates the risk through various mechanisms beyond the typical effects of cirrhosis, such as vascular fragility and impaired platelet functionality [20,21]. Although retroperitoneal hemorrhage typically occurs in response to trauma or oral anticoagulation therapy, spontaneous arterial bleeding rarely occurs [20]. Nonetheless, individuals with severe hepatic coagulopathy may develop hematomas even from minor incidents. TAE has been widely recognized for its potential to enhance survival among patients with cirrhosis who developed retroperitoneal hemorrhage [19]. Takamura et al. reported a survival rate of 50% (2/4) in patients undergoing TAE [19], while other studies have reported successful embolization in patients with end-stage liver disease with subsequent survival to discharge [22]. However, another report highlighted that despite the successful embolization in four TAE-treated patients, 75% (3/4) died within a few days. Our findings align with the challenging prognosis observed in patients with liver cirrhosis and SRH, suggesting the need for larger cohort studies owing to the limited sample used in these studies.

The incidence of retroperitoneal hematoma has been reported to range from 0.6% to 6.6% in patients undergoing therapeutic anticoagulation [23,24]. Although oral anticoagulation therapy is a common risk factor for retroperitoneal hemorrhage, our study found that eight patients (47.1%) with SRH received anticoagulation therapy. The management of life-threatening psoas hematomas and, more broadly, soft tissue hemorrhages in patients under anticoagulation therapy remains controversial. Conservative treatment is recommended for hemodynamically stable patients with no signs of ongoing bleeding; however, the efficacy of embolization or surgery in these scenarios has not been fully established. In general, hemodynamically unstable patients receiving anticoagulation therapy can be treated with arterial embolization, which is a minimally invasive method with a rapid therapeutic effect compared with surgical treatment [25]. In the study by M. Lukies et al., conservative management was reported to be successful in 13.2% of initially unstable patients, all of whom were receiving therapeutic anticoagulation therapy. They suggested that embolization in SRH may be most appropriate for patients with ongoing bleeding despite adequate reversal of anticoagulation agents or those who have not initially received anticoagulation therapy and are hemodynamically unstable [26]. However, conservative treatment is typically adequate for managing the majority of patients with mild retroperitoneal hemorrhages. The psoas muscle is capable of accumulating up to ten times its volume. Consequently, a psoas hematoma or retroperitoneal hemorrhage can present with hypotension, abdominal pain, or a decrease in hemoglobin levels. In our study, all patients (*n* = 8) under oral anticoagulation therapy who showed active bleeding on CT underwent aggressive TAE regardless of their hemodynamic status. The patients demonstrated a mean shock index of 0.98 (range: 0.47–1.53), with 50% (four out of eight) exhibiting a shock index of >1.0 and the remainder below this threshold. Notably, this strategy resulted in clinical success in 87.5% of the patients (seven out of eight), thereby indicating a favorable prognosis without major complications. Therefore, aggressive angiography and embolization are recommended in patients receiving anticoagulation therapy when CT reveals active bleeding in the retroperitoneal and/or psoas muscle areas.

In the literature, the mortality rate in patients with unstable traumatic retroperitoneal hemorrhage is approximately 17.6% to 47% [6,27]. A previous study utilized TAE to manage life-threatening bleeding resulting from blunt trauma or iatrogenic injury, achieving hemostasis in all patients, with an overall clinical success rate of 87.5%. Specifically, for bleeding originating from the lumbar artery, a clinical success rate of 80% (4/5) was reported [27]. Our study corroborated these findings in trauma patients, demonstrating a technical success rate of 100% and a clinical success rate of 84.2%, with mortality limited to only one patient (5.3%). These results are consistent with those of previous studies. However, contrasting findings were reported by another study that examined patients with lumbar artery injuries following blunt trauma [28]. Of the 21 bleeding sites identified, successful embolization was achieved in 16 (76.2%), while technical failure occurred in 2 (9.5%); meanwhile, conservative treatment was chosen for three sites (14.3%) due to concerns over an unintentional embolization of the artery of Adamkiewicz. Notably, 50% of patients (8/16) died due to severe brain injury, hypovolemic shock, or stroke and did not undergo embolization [28]. Our study likely yielded more favorable clinical outcomes as it focused on patients with retroperitoneal bleeding as the primary concern, excluding those whose condition was significantly affected by bleeding from other sites.

In our study, aside from fever, which resolved within a few days following conservative treatment in 11.1% of patients who underwent TAE, no significant complications occurred. However, one patient exhibited motor weakness in the ipsilateral lower extremity following embolization, raising suspicion of a complication involving the spinal cord, such as the Brown-Séquard syndrome (BSS). BSS is observed in patients with incomplete spinal cord injuries and affects a hemisection of the spinal cord. BSS with a vascular origin rarely occurs as the large vascular territories are not divided hemilaterally. Only a few cases of vascular-related BSS have been reported. Neurological damage due to spinal cord ischemia is an extremely rare adverse event of peripheral embolization procedures. It occurs following TAE using TAGM for a vertebral hemangioma involving the eighth and ninth intercostal arteries [29], as well as after thoracic endovascular aortic repair for aortic dissection [30]. In addition to BSS, a case similar to ours was reported by Grigoriadis et al., wherein a coronavirus disease 19-positive patient receiving anticoagulation therapy developed SRH and underwent TAE using NBCA after superselecting the lumbar artery, leading to paralysis of both lower extremities. Subsequent MRI revealed ischemia and edema in the anterior part of the spinal cord from T11 to L5 [31]. There may occasionally be a collateral network to a radiculomedullary artery originating from the lower lumbar artery, necessitating utmost caution during the procedures [31]. Thus, in patients with psoas or retroperitoneal hemorrhage, if evident bleeding from the lumbar artery is observed on angiography, superselective embolization should be carefully performed. If embolization is deemed necessary, and a connection to the radiculomedullary artery is suspected, it is crucial to superselect the target arterial branch to prevent the reflux of embolic materials and proceed with caution.

This study has some limitations. It has a small sample size and retrospective design. Furthermore, due to the small sample size, we were unable to perform separate multivariate analyses for each group. This limitation hindered our ability to thoroughly investigate potential interactions and covariate adjustments, thereby reducing the robustness and reliability of our findings. Studies with larger patient cohorts and longer follow-up periods are warranted to further validate the findings. There was heterogeneity in the patient groups and angiographic findings, with potential confounding factors. The selection of the target vessel and embolic agent varied and was dependent on the operator’s preference, which could have affected the outcomes. Some complications after the procedure may not have been detected due to the lack of a standardized follow-up protocol after TAE.

## 5. Conclusions

TAE for psoas and/or retroperitoneal hemorrhage was safe and effective, regardless of whether the bleeding was spontaneous or trauma-related. However, patients with liver cirrhosis or requiring massive transfusion due to hemorrhage may have an increased risk of clinical failure and mortality, necessitating aggressive monitoring and treatment.

## Figures and Tables

**Figure 1 jcm-13-03317-f001:**
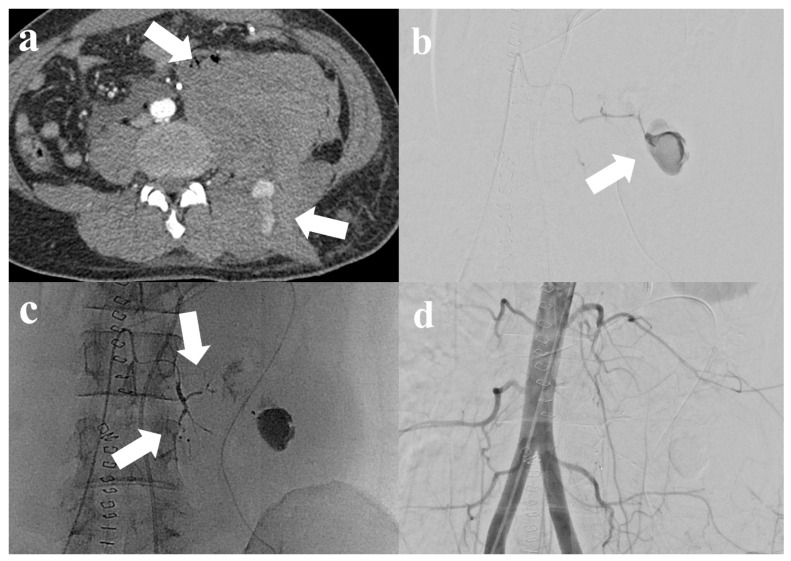
A 32-year-old male patient with blunt trauma. (**a**) Computed tomography (CT) shows huge hematoma at left retroperitoneal space and paravertebral space with contrast media extravasation (arrow). (**b**) Selective left 3rd lumbar artery angiograms show huge pseudoaneurysm (arrow), consistent with CT. (**c**) Embolization was performed using n-butyl cyanoacrylate at the left 3rd lumbar artery (arrow). (**d**) After successful embolization, abdominal aortography shows the well-embolized left 3rd lumbar artery without definite residual active bleeding.

**Figure 2 jcm-13-03317-f002:**
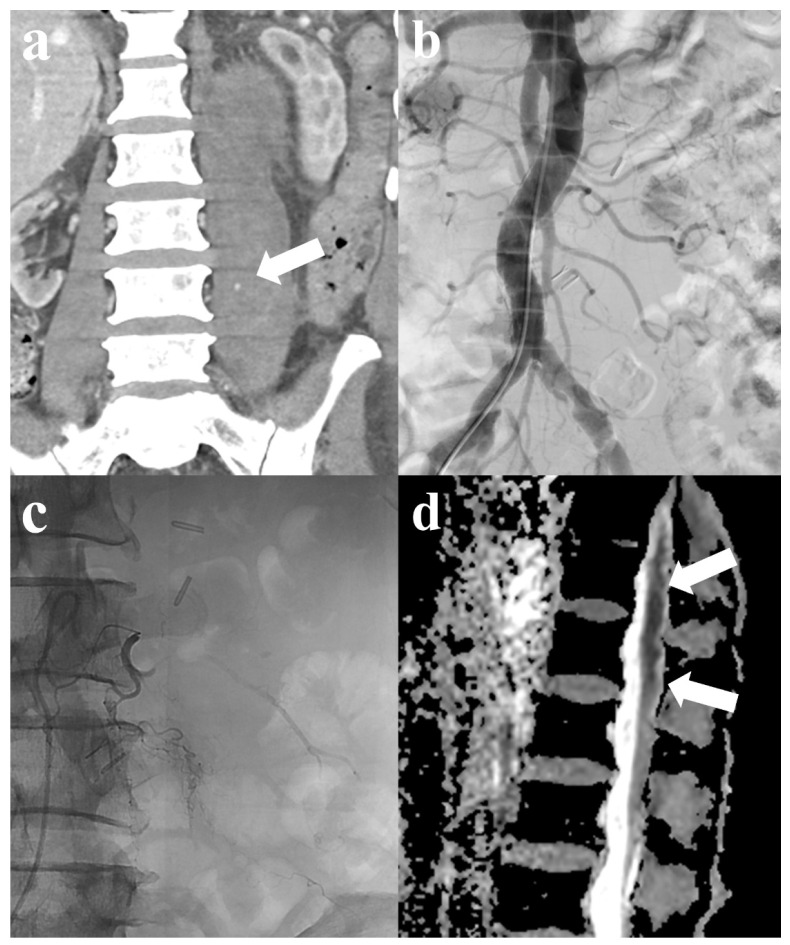
A 76-year-old male patient with spontaneous psoas hemorrhage. (**a**) Computed tomography (CT) shows contrast media extravasation with hematoma in the left psoas muscle (arrow). (**b**) Abdominal aortography reveals no definite active bleeding. Subsequent selective multilevel left lumbar artery angiography also shows no active bleeding. (**c**) Selective embolization was performed using gelatin sponge particles in the left 3rd and 4th lumbar arteries for prophylactic purposes. (**d**) After embolization, motor weakness in the ipsilateral lower extremity developed; magnetic resonance imaging (MRI) suggests spinal cord infarction, indicated by suspected diffusion restriction in the lower thoracic spinal cord through the conus medullaris (arrows).

**Table 1 jcm-13-03317-t001:** Overall group comparison between traumatic and spontaneous psoas and/or retroperitoneal hemorrhage in patients treated with transcatheter arterial embolization.

	Total Cohort (*n* = 36)	Spontaneous (*n* = 17)	Trauma (*n* = 19)	*p*
Sex (%)				0.194
Male	21 (58.3)	8 (47.1)	13 (68.4)	
Female	15 (41.7)	9 (52.9)	6 (31.6)	
Age	61.3 ± 18.2	67.0 ± 15.9	56.2 ± 19.0	0.075
Anticoagulation (%)	11 (30.6)	8 (47.1)	3 (15.8)	0.042
Underlying disease (%)				<0.001
HTN	7 (19.4)	7 (41.2)	0 (0)	
DM	4 (11.1)	4 (23.5)	0 (0)	
ESRD	4 (11.1)	3 (17.6)	1 (5.3)	
Liver cirrhosis	6 (16.7)	5 (29.4)	1 (5.3)	
CAOD	7 (19.4)	4 (23.5)	3 (15.8)	
Cerebral infarction	2 (5.6)	2 (11.8)	0 (0)	
Etc. *	5 (13.9)	3 (17.6)	2 (10.5)	
Active bleeding (%)				
CT	33 (91.7)	16 (94.1)	17 (89.5)	0.615
Angiography	25 (69.4)	10 (58.8)	15 (78.9)	0.191
Multiple	19 (52.8)	6 (35.3)	13 (68.4)	0.047
Systolic blood pressure (mmHg)	107.6 ± 28.3	118.5 ± 26.4	99.0 ± 27.4	0.045
Heart rate	99.2 ± 24.0	106.8 ± 27.7	93.3 ± 19.4	0.104
Hemoglobin (Pre, g/dL)	7.5 ± 2.1	6.6 ± 1.1	8.2 ± 2.5	0.018
Coagulation factor				
PT	15.3 ± 5.1	17.0 ± 5.9	13.8 ± 3.7	0.060
INR	1.33 ± 0.45	1.48 ± 0.52	1.19 ± 0.33	0.057
aPTT	30.6 ± 9.8	34.1 ± 10.5	28.1 ± 8.7	0.081
pRBC transfusion (unit)	4.0 ± 2.5	3.9 ± 2.7	3.8 ± 2.3	0.793
Hospital stay (day)	35.0 ± 22.5	34.4 ± 20.3	35.5 ± 24.9	0.879
Technical success (%)	36 (100)	17 (100)	19 (100)	-
Primary clinical success (%)	28 (77.8)	11 (64.7)	17 (89.5)	0.074
Major complication	1 (2.8)	1 (5.9)	0 (0)	0.284
Mortality (%)	7 (19.4)	6 (35.3)	1 (5.3)	0.023

* Etc. includes lung cancer (*n* = 2), stomach cancer (*n* = 1), neurofibromatosis (*n* = 1), and pulmonary thromboembolism and deep venous thrombosis (*n* = 1). Abbreviation: HTN—hypertension, DM—diabetes mellitus, ESRD—end-stage renal disease, CAOD—coronary artery occlusive disease, CT—computed tomography, PT—prothrombin time, INR—international normalized ratio, aPTT—activated partial thromboplastin time, pRBC—packed red blood cell.

**Table 2 jcm-13-03317-t002:** Primary clinical failure after transcatheter arterial embolization in patients with psoas and retroperitoneal hemorrhage.

Age/Sex	Underlying Disease	Group	Anticoagulation Therapy	Heart Rate per Minute	Shock Index	Active Bleeding (CT/Angiography)	Bleeding Focus and Embolized Vessel	Initial Hemoglobin (g/dL)	pRBC Transfusion (Unit)	Technical Success	Hospital Stay (Day)	Survival	Remarks
M/89	Hypertension, CAOD	SG	Y	153	1.22	Y/Y	Left 3rd lumbar artery	6.2	4	Y	16	N	Expire due to respiratory failure 7 days after embolization
F/36	Liver cirrhosis	SG	N	120	1.21	Y/Y	Right 4th lumbar artery	5.8	7	Y	24	N	
M/29	Neurofibromatosis	TG (Collision)	N	103	1.05	Y/Y	Left 1st, 3rd, and 4th lumbar artery (1st session) and left 2nd lumbar artery (2nd session)	5.1	10	Y → Y	70	Y	Clinical improvement after subsequent 2nd transarterial embolization (rebleeding from another artery and embolization was successfully performed)
F/44	Liver cirrhosis	SG	N	131	1.34	Y/Y	Left 3rd and 4th lumbar artery; deep circumflex iliac artery	4	9	Y	20	N	
M/77	None	TG (Traffic accident)	N	89	1.09	Y/Y	Both iliolumbar arteries	6.5	6	Y	29	N	
F/78	Liver cirrhosis	SG	N	71	0.72	Y/N	Both 2nd and 3rd lumbar arteries	6.6	5	Y	18	N	
M/59	Liver cirrhosis	SG	N	80	0.73	Y/N	Right 4th lumbar artery	6.8	2	Y	88	N	
F/45	Liver cirrhosis	SG	N	102-	0.77	Y/Y	Left 2nd, 3rd, and 4th lumbar artery	7.2	2	Y	32	N	

Abbreviation: CAOD—coronary arterial occlusive disease, CT—computed tomography, pRBC—packed red blood cell, SG—spontaneous group, TG—trauma group.

**Table 3 jcm-13-03317-t003:** Clinical outcome in patients with spontaneous psoas and/or retroperitoneal hemorrhage treated with transcatheter arterial embolization.

	Primary Clinical Success		
	Yes (*n* = 11)	No (*n* = 6)	*p* Value
Sex (%)			0.373
Male	6	2	
Female	5	4	
Age	71.6 ± 10.7	58.5 ± 21.0	0.191
Systolic blood pressure (mmHg)	124.6 ± 29.9	110.7 ± 15.1	0.447
Heart rate	104.7 ± 25.4	109.7 ± 31.2	0.828
Anticoagulation			0.088
Yes	7	1	
No	4	5	
Underlying (%)			0.001
Liver cirrhosis	0	5	
Non-liver cirrhosis	11	1	
Active bleeding on CT			0.647
Yes	10	6	
No	1	0	
Active bleeding on angiography			0.516
Yes	6	4	
No	5	2	
Bleeding from multiple sites			0.339
Yes	3	3	
No	8	3	
Hemoglobin (Pre, g/dL)	6.89 ± 1.04	6.10 ± 1.14	0.191
Coagulation factor			
PT	13.5 ± 1.8	23.4 ± 5.5	0.004
INR	1.18 ± 0.17	2.03 ± 0.51	0.007
aPTT	30.8 ± 9.8	42.5 ± 7.8	0.056
pRBC transfusion (unit)	3.27 ± 2.65	5.00 ± 2.76	0.183
Hospital stay (day)	35.1 ± 16.7	33.0 ± 27.5	0.291

Abbreviation: CT—computed tomography, PT—prothrombin time, INR—international normalized ratio, aPTT—activated partial thromboplastin time, pRBC—packed red blood cell.

**Table 4 jcm-13-03317-t004:** Clinical outcome in patients with traumatic psoas and/or retroperitoneal hemorrhage treated with transcatheter arterial embolization.

	Primary Clinical Success		
	Yes (*n* = 17)	No (*n* = 2)	*p* Value
Sex (%)			0.310
Male	11	2	
Female	6	0	
Age	56.6 ± 18.3	53.0 ± 33.9	0.809
Systolic blood pressure (mmHg)	100.1 ± 28.8	90.0 ± 11.3	0.638
Heart rate	92.9 ± 20.4	96.0 ± 9.9	0.840
Anticoagulation			0.517
Yes	3	0	
No	14	2	
Trauma			0.545
Traffic accident	7	1	
Fell down	6	0	
Others *	4	1	
Active bleeding on CT			0.608
Yes	15	2	
No	2	0	
Active bleeding on angiography			0.440
Yes	13	2	
No	4	0	
Bleeding from multiple sites			0.339
Yes	11	2	
No	6	0	
Hemoglobin (Pre, g/dL)	8.52 ± 2.45	5.80 ± 0.99	0.146
Coagulation factor			
PT	13.8 ± 4.0	13.7 ± 0.8	0.968
INR	1.19 ± 0.35	1.17 ± 0.10	0.926
aPTT	28.5 ± 9.1	24.5 ± 0.5	0.543
pRBC transfusion (unit)	3.65 ± 1.87	8.00 ± 2.83	0.008
Hospital stay (day)	33.9 ± 24.9	49.5 ± 29.0	0.418

* Others include stab injury (*n* = 2), collision (*n* = 1), slip down (*n* = 1), and surgery (*n* = 1). Abbreviation: CT—computed tomography, PT—prothrombin time, INR—international normalized ratio, aPTT—activated partial thromboplastin time, pRBC—packed red blood cell.

**Table 5 jcm-13-03317-t005:** Multivariable logistic regression analysis of independent factors for primary clinical failure in all patients with psoas and retroperitoneal hemorrhage.

Characteristics	Univariate		Multivariate	
	OR (95% CI)	*p*	OR (95% CI)	*p*
Sex				
Male	0.647 (0.133–3.141)	0.589		
Female				
Age	1.016 (0.974–1.061)	0.459		
Liver cirrhosis				
Yes	43.333 (3.712–505.806)	0.003	55.055 (2.439–1242.650)	0.012
No				
Trauma				
Yes	0.216 (0.037–1.267)	0.090		
No				
Anticoagulation				
Yes	0.257 (0.028–2.399)	0.233		
No				
Shock index				
≥1.0	1.548 (0.307–7.806)	0.597		
<1.0				
Hemoglobin (g/dL)				
<6.0	12.600 (1.350–117.570)	0.026	4.097 (0.228–73.658)	0.339
≥6.0				
pRBC transfusion (unit)	1.459 (1.024–2.078)	0.037	1.460 (0.844–2.526)	0.176
Multiple bleeding				
Yes	1.667 (0.333–8.352)	0.534		
No				
Active bleeding on CT				
Yes	<0.001	0.999		
No				
Active bleeding on angiography				
Yes	0.704 (0.118–4.198)	0.700		
No				
Hospital stay	0.995 (0.961–1.030)	0.756		

Abbreviation: INR—international normalized ratio, pRBC—packed red blood cell, CT—computed tomography.

## Data Availability

All data are available through the corresponding authors.

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
