# Peer review of "Clinical Outcomes and Safety of Transcatheter Arterial Embolization in Patients with Traumatic or Spontaneous Psoas and Retroperitoneal Hemorrhage"

_jcm, 2024, doi:10.3390/jcm13113317_

Round 1
Reviewer 1 Report
Comments and Suggestions for Authors
Line 71: Please clarify what you mean by "anastomoses from the primary bleeding site". As there are no anastomoses, I do not understand what is bleeding.
Line 79: I do not feel you meant to say "a final renal arteriography was performed". Please clarify what you meant to say.
Line 112: To what study are you referring when you start the sentence with "In a previous study".
Line 126: I feel that although it makes sense as there is a high n if you analyze both the SG and TG groups as as one group to determine by means of multivariate logistic regression analysis the independent risk factors, you made a strong point in Table 1 that the two groups are different. Since the only independent risk factor was the amount of blood lost, which is not very meaningful or helpful. I suggest you consider the two groups independently in the regression analysis, because I feel you would come up with more useful factors to help practioners know how best to mange those patients.
The SG has three clinical failures and only one death. In the TG, there were 7 clinical failures and 7 deaths. What does clinical failure in the SG mean, if not death?
I do not like Table 2 because it is horizontal and very difficult to read and understand. I have many questions: Column #1: How are the patients ranked? Is it by time, so that #1 was the first patient in the group with clinical failure? Please explain; Column #2: This is self explanatory; Column #3: Its title is "Underlying?" which is an odd term. Do you mean the Underlying Diagnosis? If so, please say so in both Tables 1 and 2. Were there any patients with two concerning underlying diagnoses?; Column #4: This column should be titled "Group" and each patient should have listed as being in SG or TG. ; Column #5: This column should be titled "On anticoagulation?".; Column #6: I suggest that this column should be titled "Hypotensive?", which needs to be explained as hypotensive on admission (Please add a Legend for this table explaining the columns to aid the reader in understanding what you are wanting them to know.) And for each patient, there should be a "Y" or "N".: Column #7: I am not sure this column is needed, as the important issue si whether the patients were hypotensive on admission. A lack of tachycardia could mean the patients was on adequate Beta Blockers.: Column #8: The column is fine.: Column #9: I do not understand why there are two answers. Please explain.: Column #10: The title is unclear, as it is "Bleeding focus &". I think it should be source of the bleeding in terms of the named artery(ies). So it probably should be titled as "Artery(ies) bleeding" and for each patient the name of the artery(ies) found to be bleeding should be given. For instance, the information given for Patients #1 & 2 is fine, but the arteries bleeding were not given for Patient #3.: Column #11: I assume that "Hemoglobin (Pre)" means hemoglobin on admission, which I would clarify in the Table's Legend.: Column #12: Please complete the title. I assume you meant (Units of pRBC transfused.) (Another issue is please give units where needed or in the Legend.): Column #13: I assume that the title should be "Technical success?". If so, please change it. Also what does it mean that there are two "Y"s for patient #3? And why was there technical success for Patient #7, when there was first no active bleeding. I assume that the "N/Y" for that patient in Column #9, meant that initially there was no active bleeding but later there was. That would be answered in the Table's Legend.: Column #14: I assume that "Hospital Stay" is in days. Were any of the patients readmitted?: Columns #15 & 16: The title for Column #15 should be "Survival?" I have asked previously for an explanation why Patients #3 & 7 are called Clinical Failures, yet survived but all the other "Clinical Failures" died. The usual thinking would be that a clinical failure occurred because they continued to bleed after embolization, yet those patients were listed as having a technical success and Patient #3 was listed as having two technical successes! This issue needs to be addressed. Additionaly why does Patient #1 in Column #16 have a remark, which is not grammatical of "Expire due to?" and none of the others do? I suggest eliminating Column #16, and putting an asterix on the Y for Patients #3 & 7, which would be explained in the Table's Legend.
Comments on the Quality of English LanguageLine 30: add an "and" before "sensorimotor".
Line 38: add an "is" after "blunt trauma".
Line 45: add "hemorrhage" after "space".
Line 81: Replace "at" with "in".
Line 84: Replace "at left" with "of left".
In the last line of Table 1, Mortality is misspelt.
Line 147: Place a "with" between "associated" and "primary".
Line 175: I feel "arteries" is meant and not "supplies".
Line 200: I suggest rewriting the sentence from "This increase in the mortality rate" to "The higher mortality rate".
Line 202: I suggest eliminating "starkly".
Line 245: Replace "under" to "on".
Author Response
Responses to the Associate Editor’s and Reviewers’ Comments
26 May, 2024
Dear reviewers and editorial staffs in Journal of Clinical Medicine
We are sincerely grateful for your thorough consideration and scrutiny of our manuscript, “Clinical Outcomes and Safety of Transcatheter Arterial Embolization in Patients with Traumatic or Spontaneous Psoas and Retroperitoneal Hemorrhage”. Through the accurate comments made by the editors and reviewers, we better understand the critical issues in this paper. We have revised the manuscript according to the editor’s and reviewer’s suggestions. We hope that our revised manuscript will be considered and accepted for publication in the Journal of Clinical Medicine. We acknowledge that the scientific and clinical quality of our manuscript was improved by the scrutinizing efforts of the reviewers and editors.
The changes within the revised manuscript_Annotated were highlighted (underlined in blue). Point-by-point responses to the editor’s comments are provided below.
Sincerely,
Chang Hoon Oh.
----------------------------------------------------------------------------------------------------------
Reviewer 1.
Reviewer’s comment:
Line 71: Please clarify what you mean by "anastomoses from the primary bleeding site". As there are no anastomoses, I do not understand what is bleeding.
Author’s Response:
We appreciate the comment. We changed 'anastomoses from the primary bleeding site' to 'blood flow through collaterals' because anastomoses might be confusing, and it better describes the bleeding from the primary site through collateral flow.
blood flow through collaterals from the primary bleeding site
Reviewer’s comment:
Line 79: I do not feel you meant to say "a final renal arteriography was performed". Please clarify what you meant to say.
Author’s Response:
We appreciate the comment. Based on your comment, it seems the meaning might be confusing. It was intended to refer to the post-embolization angiography for confirmation after embolization, so I have revised it as follows.
final renal arteriography after embolization
Reviewer’s comment:
Line 112: To what study are you referring when you start the sentence with "In a previous study".
Author’s Response:
We appreciate the comment. This refers to the current study, meaning 'present study,' which was mistakenly written as 'previous.' I have corrected it as follows. I apologize for the basic error.
In a previous present study
Reviewer’s comment:
Line 126: I feel that although it makes sense as there is a high n if you analyze both the SG and TG groups as as one group to determine by means of multivariate logistic regression analysis the independent risk factors, you made a strong point in Table 1 that the two groups are different. Since the only independent risk factor was the amount of blood lost, which is not very meaningful or helpful. I suggest you consider the two groups independently in the regression analysis, because I feel you would come up with more useful factors to help practioners know how best to mange those patients.
Author’s Response:
We appreciate the comment. Originally, we intended to perform a multivariate analysis separately for the SG and TG groups. However, due to the small sample size in each group, the statistical reliability was compromised, leading us to combine the two groups for analysis. Instead, to preserve the original research plan for assessing clinical success and failure within each group, we conducted Fisher's exact test analysis for SG and TG, respectively, which is added in Tables 3 and 4. The corresponding values and explanations have also been included in the results section. Thank you.
In 17 patients SG, 6 patients experienced clinical failure. Among the patients SG with liver cirrhosis, there were a total of 6 patients, 5 of whom experienced clinical failure (83.3%, P = 0.001). Additionally, prothrombin time (13.5 ± 1.8 vs. 23.4 ± 5.5, P = 0.004) and INR (1.18 ± 0.17 vs. 2.03 ± 0.51, P = 0.007) were statistically significant (Table 3). Among the 19 TG patients, there were 2 cases of clinical failure. Among the variables, only pRBC transfusion was significant (clinical success: 3.65 ± 1.87 vs. clinical failure: 8.00 ± 2.83; P = 0.008) (Table 4).
Table 3. Clinical outcome in patients with spontaneous psoas and/or retroperitoneal hemorrhage treated transcatheter arterial embolization
|
|
Clinical success |
|
|
|
|
Yes (n = 11) |
No (n = 6) |
P value |
|
Sex (%) |
|
|
0.373 |
|
Male |
6 |
2 |
|
|
Female |
5 |
4 |
|
|
Age |
71.6 10.7 |
58.5 21.0 |
0.191 |
|
Systolic blood pressure (mmHg) |
124.6 29.9 |
110.7 15.1 |
0.447 |
|
Heart rate |
104.7 25.4 |
109.7 31.2 |
0.828 |
|
Anticoagulation |
|
|
0.088 |
|
Yes |
7 |
1 |
|
|
No |
4 |
5 |
|
|
Underlying (%) |
|
|
0.001 |
|
Liver cirrhosis |
0 |
5 |
|
|
Non-liver cirrhosis |
11 |
1 |
|
|
Active bleeding on CT |
|
|
0.647 |
|
Yes |
10 |
6 |
|
|
No |
1 |
0 |
|
|
Active bleeding on angiography |
|
|
0.516 |
|
Yes |
6 |
4 |
|
|
No |
5 |
2 |
|
|
Bleeding from multiple site |
|
|
0.339 |
|
Yes |
3 |
3 |
|
|
No |
8 |
3 |
|
|
Hemoglobin (Pre, g/dL) |
6.89 1.04 |
6.10 1.14 |
0.191 |
|
Coagulation factor |
|
|
|
|
PT |
13.5 1.8 |
23.4 5.5 |
0.004 |
|
INR |
1.18 0.17 |
2.03 0.51 |
0.007 |
|
aPTT |
30.8 9.8 |
42.5 7.8 |
0.056 |
|
pRBC transfusion (unit) |
3.27 2.65 |
5.00 2.76 |
0.183 |
|
Hospital stay (day) |
35.1 16.7 |
33.0 27.5 |
0.291 |
* CT computed tomography, PT prothrombin time, INR international normalized ratio, aPTT activated partial thromboplastin time, pRBC packed red blood cell
Table 4. Clinical outcome in patients with traumatic psoas and/or retroperitoneal hemorrhage treated transcatheter arterial embolization
|
|
Clinical success |
|
|
|
|
Yes (n = 17) |
No (n = 2) |
P value |
|
Sex (%) |
|
|
0.310 |
|
Male |
11 |
2 |
|
|
Female |
6 |
0 |
|
|
Age |
56.6 18.3 |
53.0 33.9 |
0.809 |
|
Systolic blood pressure (mmHg) |
100.1 28.8 |
90.0 11.3 |
0.638 |
|
Heart rate |
92.9 20.4 |
96.0 9.9 |
0.840 |
|
Anticoagulation |
|
|
0.517 |
|
Yes |
3 |
0 |
|
|
No |
14 |
2 |
|
|
Trauma |
|
|
0.545 |
|
Traffic accident |
7 |
1 |
|
|
Fall down |
6 |
0 |
|
|
Others* |
4 |
1 |
|
|
Active bleeding on CT |
|
|
0.608 |
|
Yes |
15 |
2 |
|
|
No |
2 |
0 |
|
|
Active bleeding on angiography |
|
|
0.440 |
|
Yes |
13 |
2 |
|
|
No |
4 |
0 |
|
|
Bleeding from multiple site |
|
|
0.339 |
|
Yes |
11 |
2 |
|
|
No |
6 |
0 |
|
|
Hemoglobin (Pre, g/dL) |
8.52 2.45 |
5.80 0.99 |
0.146 |
|
Coagulation factor |
|
|
|
|
PT |
13.8 4.0 |
13.7 0.8 |
0.968 |
|
INR |
1.19 0.35 |
1.17 0.10 |
0.926 |
|
aPTT |
28.5 9.1 |
24.5 0.5 |
0.543 |
|
pRBC transfusion (unit) |
3.65 1.87 |
8.00 2.83 |
0.008 |
|
Hospital stay (day) |
33.9 24.9 |
49.5 29.0 |
0.418 |
* Others includes Stab injury (n = 2), collision (n = 1), slip down (n = 1), surgery (n = 1).
** CT computed tomography, PT prothrombin time, INR international normalized ratio, aPTT activated partial thromboplastin time, pRBC packed red blood cell
Reviewer’s comment:
The SG has three clinical failures and only one death. In the TG, there were 7 clinical failures and 7 deaths. What does clinical failure in the SG mean, if not death?
Author’s Response:
We appreciate the comment. As shown in Tables 1 and 2, there were 6 clinical failures in the SG (spontaneous group), and all of these patients expired. In the TG (trauma group), there were 3 clinical failures, but 1 patient improved after undergoing TAE once more, another improved after a few days of conservative treatment (following 5 pRBC transfusions), and the remaining patient expired. In the Methods section under Study Endpoints, it is mentioned that ‘Clinical success was defined as the resolution of the signs and symptoms of bleeding with no requirement for interventional or surgical hemostasis.’ Therefore, we will consider the 1 patient who expired and the 1 patient who required reintervention as clinical failures, and the patient who improved after 5 pRBC transfusions as a clinical success. Accordingly, the clinical success in TG has been updated to 17 patients, and the total number of clinical successes has been updated to 28 patients. Consequently, the statistics were recalculated, and the p-value was obtained again. All relevant content and values related to clinical failure in the manuscript have been updated, and as a result, the values in the multivariate logistic regression analysis have also changed. Please review the updated figures and content carefully. An explanation for this has also been added to the results section as follows.
Among the total of 8 patients who experienced clinical failure, one patient with Neurofibromatosis suffered hemorrhage after a collision. This patient underwent a second session of TAE due to unstable vital signs following the first TAE. During the second session, another arterial bleeding site that had not previously been identified was discovered and successfully embolized, leading to clinical improvement. The remaining 7 patients (87.5%) expired. (Table 2).
Reviewer’s comment:
I do not like Table 2 because it is horizontal and very difficult to read and understand. I have many questions: Column #1: How are the patients ranked? Is it by time, so that #1 was the first patient in the group with clinical failure? Please explain; Column #2: This is self explanatory; Column #3: Its title is "Underlying?" which is an odd term. Do you mean the Underlying Diagnosis? If so, please say so in both Tables 1 and 2. Were there any patients with two concerning underlying diagnoses?; Column #4: This column should be titled "Group" and each patient should have listed as being in SG or TG. ; Column #5: This column should be titled "On anticoagulation?".; Column #6: I suggest that this column should be titled "Hypotensive?", which needs to be explained as hypotensive on admission (Please add a Legend for this table explaining the columns to aid the reader in understanding what you are wanting them to know.) And for each patient, there should be a "Y" or "N".: Column #7: I am not sure this column is needed, as the important issue si whether the patients were hypotensive on admission. A lack of tachycardia could mean the patients was on adequate Beta Blockers.: Column #8: The column is fine.: Column #9: I do not understand why there are two answers. Please explain.: Column #10: The title is unclear, as it is "Bleeding focus &". I think it should be source of the bleeding in terms of the named artery(ies). So it probably should be titled as "Artery(ies) bleeding" and for each patient the name of the artery(ies) found to be bleeding should be given. For instance, the information given for Patients #1 & 2 is fine, but the arteries bleeding were not given for Patient #3.: Column #11: I assume that "Hemoglobin (Pre)" means hemoglobin on admission, which I would clarify in the Table's Legend.: Column #12: Please complete the title. I assume you meant (Units of pRBC transfused.) (Another issue is please give units where needed or in the Legend.): Column #13: I assume that the title should be "Technical success?". If so, please change it. Also what does it mean that there are two "Y"s for patient #3? And why was there technical success for Patient #7, when there was first no active bleeding. I assume that the "N/Y" for that patient in Column #9, meant that initially there was no active bleeding but later there was. That would be answered in the Table's Legend.: Column #14: I assume that "Hospital Stay" is in days. Were any of the patients readmitted?: Columns #15 & 16: The title for Column #15 should be "Survival?" I have asked previously for an explanation why Patients #3 & 7 are called Clinical Failures, yet survived but all the other "Clinical Failures" died. The usual thinking would be that a clinical failure occurred because they continued to bleed after embolization, yet those patients were listed as having a technical success and Patient #3 was listed as having two technical successes! This issue needs to be addressed. Additionaly why does Patient #1 in Column #16 have a remark, which is not grammatical of "Expire due to?" and none of the others do? I suggest eliminating Column #16, and putting an asterix on the Y for Patients #3 & 7, which would be explained in the Table's Legend.
Author’s Response:
We appreciate the comment. Due to the significant and important points raised in the comments, I have revised the tables as follows. Patient numbers were initially arranged in reverse chronological order based on the procedures performed at our hospital, focusing only on those with primary clinical failure, which might have caused confusion, hence they were removed. Additionally, while transferring the tables vertically in Word to PDF, some content was cut off, so I adjusted the overall table size. Following the comments, I changed to TG or SG, and noted the trauma mechanism for TG patients within parentheses. 'Anticoagulation' was also modified to 'anticoagulation therapy'. Since the shock index is the ratio of heart rate to systolic pressure, and many patients had systolic pressures between 90-100, potentially underestimating hypotension, I removed systolic blood pressure and retained only heart rate and shock index for clarity. Furthermore, the third patient underwent two TAEs; the first TAE successfully embolized a bleeding arterial branch, but hemodynamic instability persisted, leading to another TAE where another previously non-bleeding arterial branch was embolized. Although both were technical successes, I classified them as primary clinical failure in our study because our criteria for primary clinical success defined it as recovery without additional surgical or interventional procedures post-TAE. To avoid confusion, I have changed 'clinical success or failure' to 'primary clinical success or failure' to clarify that clinical failure does not necessarily mean death. In response to inquiries about whether it is appropriate to classify a case as a technical success when active bleeding is not visible on angiography, I have added further clarification to the methods section. Specifically, 'Additionally, in cases where CME or pseudoaneurysm was not clearly identifiable on angiography, occlusion was performed on the vessel supplying the same anatomical territory surrounding the hematoma.' Even if angiography is negative, I believe it should be considered a technical success when prophylactic embolization is performed on the suspected target vessel.
Table 2. Primary clinical failure patients after transcatheter arterial embolization in patients with psoas and retroperitoneal hemorrhage
|
9 |
8 |
7 |
6 |
5 |
4 |
3 |
2 |
1 |
Patient No. |
|
F/45 |
M/59 |
M/62 |
F/78 |
M/77 |
F/44 |
M/29 |
F/36 |
M/89 |
Age/Sex |
|
Liver cirrhosis |
Liver cirrhosis |
none |
Liver cirrhosis |
none |
Liver cirrhosis |
Neurofibromatosis |
Liver cirrhosis |
Hypertension, CAOD |
Underlying disease |
|
SG |
SG |
Fall down |
SG |
TG (Traffic accident) |
SG |
TG (Collision) |
SG |
SG |
Group |
|
N |
N |
N |
N |
N |
N |
N |
N |
Y |
Anticoagulation therapy |
|
133 |
110 |
90 |
99 |
82 |
98 |
98 |
99 |
125 |
Systolic blood pressure (mmHg) |
|
102- |
80 |
82 |
71 |
89 |
131 |
103 |
120 |
153 |
Heart rate per minute |
|
0.77 |
0.73 |
0.91 |
0.72 |
1.09 |
1.34 |
1.05 |
1.21 |
1.22 |
Shock index |
|
Y / Y |
Y / N |
N / Y |
Y / N |
Y / Y |
Y / Y |
Y / Y |
Y / Y |
Y / Y |
Active bleeding |
|
Left 2nd, 3rd, 4th lumbar artery |
Right 4th lumbar artery |
Right iliolumbar artery, posterior division of internal iliac artery |
Both 2nd, 3rd lumbar artery |
Both iliolumbar artery |
Left 3rd, 4th lumbar artery, Deep circumflex iliac artery |
Left 1st, 3rd, 4th lumbar artery à Left 2nd lumbar artery (2nd session) |
Right 4th lumbar artery |
Left 3rd lumbar artery |
Bleeding focus & embolized vessel |
|
7.2 |
6.8 |
10.2 |
6.6 |
6.5 |
4.0 |
5.1 |
5.8 |
6.2 |
Initial hemoglobin (g/dL) |
|
2 |
2 |
7 |
5 |
6 |
9 |
10 |
7 |
4 |
pRBC transfusion (unit) |
|
Y |
Y |
Y |
Y |
Y |
Y |
Y à Y |
Y |
Y |
Technical success |
|
32 |
88 |
28 |
18 |
29 |
20 |
70 |
24 |
16 |
Hospital stay (day) |
|
N |
N |
Y |
N |
N |
N |
Y |
N |
N |
Survival |
|
|
|
Clinical improvement after conservative treatment (5 pRBC) |
|
|
|
Clinical improvement after subsequent 2nd transarterial embolization (rebleeding from another artery and embolization was successfully performed) |
|
Expire due to respiratory failure 7 days after embolization |
Remarks |
* CAOD coronary arterial occlusive disease, CT computed tomography, pRBC packed red blood cell, SG Spontaneous group, TG Trauma group
Once again, thank you so much your delicate comments.

Reviewer 2 Report
Comments and Suggestions for Authors
The work presents several cases treated in a similar way over the years. It presents concise results that are relevant to medical literature. It is well written and described, I suggest approval in the form it is presented.
Author Response
Responses to the Associate Editor’s and Reviewers’ Comments
26 May, 2024
Dear reviewers and editorial staffs in Journal of Clinical Medicine
We are sincerely grateful for your thorough consideration and scrutiny of our manuscript, “Clinical Outcomes and Safety of Transcatheter Arterial Embolization in Patients with Traumatic or Spontaneous Psoas and Retroperitoneal Hemorrhage”. Through the accurate comments made by the editors and reviewers, we better understand the critical issues in this paper. We have revised the manuscript according to the editor’s and reviewer’s suggestions. We hope that our revised manuscript will be considered and accepted for publication in the Journal of Clinical Medicine. We acknowledge that the scientific and clinical quality of our manuscript was improved by the scrutinizing efforts of the reviewers and editors.
The changes within the revised manuscript_Annotated were highlighted (underlined in blue). Point-by-point responses to the editor’s comments are provided below.
Sincerely,
Chang Hoon Oh.
----------------------------------------------------------------------------------------------------------
Reviewer 2.
Reviewer’s comment:
The work presents several cases treated in a similar way over the years. It presents concise results that are relevant to medical literature. It is well written and described, I suggest approval in the form it is presented.
Author’s Response:
We appreciate the comment. Thank you for the good review. I am so happy to be able to publish this paper in Journal of Clinical Medicine. I will continue to work hard to conduct more research that can be of great help to Journal of Clinical Medicine and publish it.
Once again, thank you so much your delicate comments.
Reviewer 3 Report
Comments and Suggestions for Authors
Congratulations for your interesting work regarding TAE. Your manuscript is well-organized and written. The content is comprehensive. Some suggestions for improvement:
- In the abstract you mention that your included patients are with cancer. You do not mention the same thing in the main body of your manuscript. If your patients are patients with cancer, you should mention that clearly in the Methods part, as this fact could affect your results. As a consequence, you should mention this limitation too in the respective paragraph of your discussion.
- You also mention that you included patients aged 10-90 years with traumatic hemorrhage, separately from the initial mentioning of the 36 patients included. I suppose you mean that these patients are a subgroup included in your study. Please make clearer you inclusion criteria.
- Regarding the platelet count which is the lower limit for inclusion of your patients, I suppose that you mean 50.000/μL. It seems that there is not the 4th power after number 10 and it seems that you mean 520/μL.
- Most of your patient with worse outcomes were patients with liver cirrhosis and these patients regard the spontaneous hemorrhage group. Obviously and as you mention these patients were end-stage liver disease patients. By taking into account that liver cirrhosis in the end-stage causes extreme coagulation disorders, shouldn't you exclude these patients in order to exclude this bias? Comparing mortality and rest results btween the two patients groups by including these patients in the one group exclusively, decreases the reliability of your results. If these patinets were included in both groups in a more balanced way I could accept the results more easily. I think that this is a methodological error. Otherwise, please present the results of your study as one or two separate patient groups by mentioning this comorbidity, but without trying to compare the two groups.
Author Response
Responses to the Associate Editor’s and Reviewers’ Comments
26 May, 2024
Dear reviewers and editorial staffs in Journal of Clinical Medicine
We are sincerely grateful for your thorough consideration and scrutiny of our manuscript, “Clinical Outcomes and Safety of Transcatheter Arterial Embolization in Patients with Traumatic or Spontaneous Psoas and Retroperitoneal Hemorrhage”. Through the accurate comments made by the editors and reviewers, we better understand the critical issues in this paper. We have revised the manuscript according to the editor’s and reviewer’s suggestions. We hope that our revised manuscript will be considered and accepted for publication in the Journal of Clinical Medicine. We acknowledge that the scientific and clinical quality of our manuscript was improved by the scrutinizing efforts of the reviewers and editors.
The changes within the revised manuscript_Annotated were highlighted (underlined in blue). Point-by-point responses to the editor’s comments are provided below.
Sincerely,
Chang Hoon Oh.
----------------------------------------------------------------------------------------------------------
Reviewer 3.
Reviewer’s comment:
Congratulations for your interesting work regarding TAE. Your manuscript is well-organized and written. The content is comprehensive. Some suggestions for improvement:
- In the abstract you mention that your included patients are with cancer. You do not mention the same thing in the main body of your manuscript. If your patients are patients with cancer, you should mention that clearly in the Methods part, as this fact could affect your results. As a consequence, you should mention this limitation too in the respective paragraph of your discussion.
Author’s Response:
We appreciate the comment. I reviewed it again based on the comment and deleted the part as it was incorrect. Thank you.
Reviewer’s comment:
- You also mention that you included patients aged 10-90 years with traumatic hemorrhage, separately from the initial mentioning of the 36 patients included. I suppose you mean that these patients are a subgroup included in your study. Please make clearer you inclusion criteria.
Author’s Response:
We appreciate the comment. In the methods section, we included patients aged 10-90 years, with or without traumatic injury, thus encompassing both those with bleeding due to trauma and those with non-traumatic bleeding. If the meaning appears unclear, I am willing to revise it accordingly. Thank you.
Patients aged 10–90 years who underwent TAE owing to suspicion of active bleeding in the psoas or retroperitoneal space, with or without traumatic injury, were included in the study.
Reviewer’s comment:
- Regarding the platelet count which is the lower limit for inclusion of your patients, I suppose that you mean 50.000/μL. It seems that there is not the 4th power after number 10 and it seems that you mean 520/μL.
Author’s Response:
We appreciate the comment. Based on the comment, I have revised the section on platelets as follows.
abnormal platelet count (<5 × 1044 /μL)
Reviewer’s comment:
- Most of your patient with worse outcomes were patients with liver cirrhosis and these patients regard the spontaneous hemorrhage group. Obviously and as you mention these patients were end-stage liver disease patients. By taking into account that liver cirrhosis in the end-stage causes extreme coagulation disorders, shouldn't you exclude these patients in order to exclude this bias? Comparing mortality and rest results btween the two patients groups by including these patients in the one group exclusively, decreases the reliability of your results. If these patinets were included in both groups in a more balanced way I could accept the results more easily. I think that this is a methodological error. Otherwise, please present the results of your study as one or two separate patient groups by mentioning this comorbidity, but without trying to compare the two groups.
Author’s Response:
We appreciate the comment. Originally, we intended to perform a multivariate analysis separately for the SG and TG groups. However, due to the small sample size in each group, the statistical reliability was compromised, leading us to combine the two groups for analysis. Instead, to preserve the original research plan for assessing clinical success and failure within each group, we conducted Fisher's exact test analysis for SG and TG, respectively, which is added in Tables 3 and 4. The corresponding values and explanations have also been included in the results section. Thank you.
In 17 patients SG, 6 patients experienced clinical failure. Among the patients SG with liver cirrhosis, there were a total of 6 patients, 5 of whom experienced clinical failure (83.3%, P = 0.001). Additionally, prothrombin time (13.5 ± 1.8 vs. 23.4 ± 5.5, P = 0.004) and INR (1.18 ± 0.17 vs. 2.03 ± 0.51, P = 0.007) were statistically significant (Table 3). Among the 19 TG patients, there were 2 cases of clinical failure. Among the variables, only pRBC transfusion was significant (clinical success: 3.65 ± 1.87 vs. clinical failure: 8.00 ± 2.83; P = 0.008) (Table 4).
Table 3. Clinical outcome in patients with spontaneous psoas and/or retroperitoneal hemorrhage treated transcatheter arterial embolization
|
|
Clinical success |
|
|
|
|
Yes (n = 11) |
No (n = 6) |
P value |
|
Sex (%) |
|
|
0.373 |
|
Male |
6 |
2 |
|
|
Female |
5 |
4 |
|
|
Age |
71.6 10.7 |
58.5 21.0 |
0.191 |
|
Systolic blood pressure (mmHg) |
124.6 29.9 |
110.7 15.1 |
0.447 |
|
Heart rate |
104.7 25.4 |
109.7 31.2 |
0.828 |
|
Anticoagulation |
|
|
0.088 |
|
Yes |
7 |
1 |
|
|
No |
4 |
5 |
|
|
Underlying (%) |
|
|
0.001 |
|
Liver cirrhosis |
0 |
5 |
|
|
Non-liver cirrhosis |
11 |
1 |
|
|
Active bleeding on CT |
|
|
0.647 |
|
Yes |
10 |
6 |
|
|
No |
1 |
0 |
|
|
Active bleeding on angiography |
|
|
0.516 |
|
Yes |
6 |
4 |
|
|
No |
5 |
2 |
|
|
Bleeding from multiple site |
|
|
0.339 |
|
Yes |
3 |
3 |
|
|
No |
8 |
3 |
|
|
Hemoglobin (Pre, g/dL) |
6.89 1.04 |
6.10 1.14 |
0.191 |
|
Coagulation factor |
|
|
|
|
PT |
13.5 1.8 |
23.4 5.5 |
0.004 |
|
INR |
1.18 0.17 |
2.03 0.51 |
0.007 |
|
aPTT |
30.8 9.8 |
42.5 7.8 |
0.056 |
|
pRBC transfusion (unit) |
3.27 2.65 |
5.00 2.76 |
0.183 |
|
Hospital stay (day) |
35.1 16.7 |
33.0 27.5 |
0.291 |
* CT computed tomography, PT prothrombin time, INR international normalized ratio, aPTT activated partial thromboplastin time, pRBC packed red blood cell
Table 4. Clinical outcome in patients with traumatic psoas and/or retroperitoneal hemorrhage treated transcatheter arterial embolization
|
|
Clinical success |
|
|
|
|
Yes (n = 17) |
No (n = 2) |
P value |
|
Sex (%) |
|
|
0.310 |
|
Male |
11 |
2 |
|
|
Female |
6 |
0 |
|
|
Age |
56.6 18.3 |
53.0 33.9 |
0.809 |
|
Systolic blood pressure (mmHg) |
100.1 28.8 |
90.0 11.3 |
0.638 |
|
Heart rate |
92.9 20.4 |
96.0 9.9 |
0.840 |
|
Anticoagulation |
|
|
0.517 |
|
Yes |
3 |
0 |
|
|
No |
14 |
2 |
|
|
Trauma |
|
|
0.545 |
|
Traffic accident |
7 |
1 |
|
|
Fall down |
6 |
0 |
|
|
Others* |
4 |
1 |
|
|
Active bleeding on CT |
|
|
0.608 |
|
Yes |
15 |
2 |
|
|
No |
2 |
0 |
|
|
Active bleeding on angiography |
|
|
0.440 |
|
Yes |
13 |
2 |
|
|
No |
4 |
0 |
|
|
Bleeding from multiple site |
|
|
0.339 |
|
Yes |
11 |
2 |
|
|
No |
6 |
0 |
|
|
Hemoglobin (Pre, g/dL) |
8.52 2.45 |
5.80 0.99 |
0.146 |
|
Coagulation factor |
|
|
|
|
PT |
13.8 4.0 |
13.7 0.8 |
0.968 |
|
INR |
1.19 0.35 |
1.17 0.10 |
0.926 |
|
aPTT |
28.5 9.1 |
24.5 0.5 |
0.543 |
|
pRBC transfusion (unit) |
3.65 1.87 |
8.00 2.83 |
0.008 |
|
Hospital stay (day) |
33.9 24.9 |
49.5 29.0 |
0.418 |
* Others includes Stab injury (n = 2), collision (n = 1), slip down (n = 1), surgery (n = 1).
** CT computed tomography, PT prothrombin time, INR international normalized ratio, aPTT activated partial thromboplastin time, pRBC packed red blood cell

Round 2
Reviewer 3 Report
Comments and Suggestions for Authors
Congratulations again for your work. Most changes asked by the reviewers have been made, although I believe that there are some ambiguities in the manuscript caused by the small sample size Please make sure that you mention properly all the limitations of your study (small sample size, lack of multivariate analysis, which could lead to biased results, etc.
Author Response
Responses to the Associate Editor’s and Reviewers’ Comments
30 May, 2024
Dear reviewers and editorial staffs in Journal of Clinical Medicine
We are sincerely grateful for your thorough consideration and scrutiny of our manuscript, “Clinical Outcomes and Safety of Transcatheter Arterial Embolization in Patients with Traumatic or Spontaneous Psoas and Retroperitoneal Hemorrhage”. Through the accurate comments made by the editors and reviewers, we better understand the critical issues in this paper. We have revised the manuscript according to the editor’s and reviewer’s suggestions. We hope that our revised manuscript will be considered and accepted for publication in the Journal of Clinical Medicine. We acknowledge that the scientific and clinical quality of our manuscript was improved by the scrutinizing efforts of the reviewers and editors.
The changes within the revised manuscript_Annotated were highlighted (underlined in blue). Point-by-point responses to the editor’s comments are provided below.
Sincerely,
Chang Hoon Oh.
----------------------------------------------------------------------------------------------------------
Reviewer 3.
Reviewer’s comment:
Congratulations again for your work. Most changes asked by the reviewers have been made, although I believe that there are some ambiguities in the manuscript caused by the small sample size Please make sure that you mention properly all the limitations of your study (small sample size, lack of multivariate analysis, which could lead to biased results, etc.
Author’s Response:
We appreciate the comment. In response to the comments, the following content has been added to the limitations section.
Furthermore, due to the small sample size, we were unable to perform separate multivariate analyses for each group. This limitation hindered our ability to thoroughly investigate potential interactions and covariate adjustments, thereby reducing the robustness and reliability of our findings.
Once again, thank you so much your delicate comments.
